neuroscience/psychology/cognition

semantic cognition, cognitive ageing, executive function, knowledge

**Author for correspondence:**
Wei Wu
e-mail: wei.wu@ed.ac.uk

# Validated measures of semantic knowledge and semantic control: normative data from young and older adults for more than 300 semantic judgements

## Wei Wu and Paul Hoffman

School of Philosophy, Psychology and Language Sciences, University of Edinburgh, 7 George Square, Edinburgh EH8 9JZ, UK

(iD) WW, 0000-0002-0000-0858; PH, 0000-0002-3248-3225

Recent studies suggest that knowledge representations and control processes are the two key components underpinning semantic cognition, and are also crucial indicators of the shifting cognitive architecture of semantics in later life. Although there are many standardized assessments that provide measures of the quantity of semantic knowledge participants possess, normative data for tasks that probe semantic control processes are not yet available. Here, we present normative data from more than 200 young and older participants on a large set of stimuli in two semantic tasks, which probe controlled semantic processing (feature-matching task) and semantic knowledge (synonym judgement task). We verify the validity of our norms by replicating established age- and psycholinguistic-property-related effects on semantic cognition. Specifically, we find that older people have more detailed semantic knowledge than young people but have less effective semantic control processes. We also obtain expected effects of word frequency and inter-item competition on performance. Parametrically varied difficulty levels are defined for half of the stimuli based on participants' behavioural performance, allowing future studies to produce customized sets of experimental stimuli based on our norms. We provide all stimuli, data and code used for analysis, in the hope that they are useful to other researchers.

# 1. Introduction

Semantic cognition is a fundamental cognitive ability, which shapes our understanding of the environment and allows us to use our

knowledge of the world to produce context- and time-appropriate behaviour [1–4]. Neuropsychological and neuroimaging studies indicate that successful semantic cognition draws on two interacting components: semantic knowledge representations and semantic control processes [3,4]. Representation of *semantic knowledge* refers to our ability to store a wealth of information about the meanings of objects, concepts and words. In addition, *semantic control* processes regulate how we retrieve and use aspects of our knowledge in a context- and task-relevant way, e.g. by resolving competition between active representations [2,5]. All semantic tasks require the interaction of knowledge and control processes, so no task can act as a pure measure of either component. However, different tasks place greater demands on different components. Many standardized assessments of the semantic knowledge component exist, typically taking the form of vocabulary tests in which participants define words, identify words or select their synonyms (e.g. the vocabulary subtest of the WAIS or the WAIS–R, the Mill Hill vocabulary scale, and the Spot-the-Word test) [6–9]. These tests provide researchers with measures of the quantity of semantic knowledge participants possess. By contrast, we are not aware of any studies that provide normative data for tasks that probe semantic control processes, despite the critical role of these control processes in effective semantic cognition. In the present study, we provide normative data for a large set of stimuli in two semantic tasks: a feature-matching task that places high demands on controlled semantic processing, as well as a synonym judgement task that primarily assesses level of semantic knowledge.

Tests that dissociate the knowledge and the control components of semantics are particularly important for studies of cognitive ageing. It is well established that healthy individuals show age-related declines in cognitive ability across a wide variety of domains, including episodic memory, attention and processing speed, while aspects of semantic cognition are generally maintained with age [10–15]. Nevertheless, recent studies indicate that performance on semantic tasks does not show a uniform pattern of age-related changes. In fact, although semantic knowledge representations accumulate throughout the lifespan and are relatively preserved in old age, older people show deficits in tasks that place high demands on semantic control, e.g. by inducing competition between semantic representations or by requiring flexible retrieval of less salient aspects of knowledge [16–18]. These novel findings are consistent with an existing theoretical view (the default-executive coupling hypothesis of ageing, DECHA), which suggests that there is a shift in cognitive processing toward greater reliance on semantic knowledge and less on cognitive control in older adulthood [19,20]. These emerging findings indicate that assessments that probe both semantic knowledge and semantic control are essential to investigate ageing-related shifts in the cognitive architecture of semantics. Insights gained from such investigations could have wider implications for cognitive ageing in other domains.

How have the two components of semantics been assessed? As illustrated before, the semantic knowledge component is most often indexed by participants' scores on vocabulary ability tests [10–14]. For example, in the Mill Hill vocabulary scale [7], participants are asked to select the synonyms of particular words from several alternatives; in the Spot-the-Word test, participants are presented with a real word and a non-word and are asked to select the real word [6]. These vocabulary measures test participants' familiarity with words ranging in frequency from common to extremely rare (e.g. *exiguous*). Better performance indicates greater familiarity with uncommon words, which suggests that participants have a broader and richer semantic knowledge store.

The assessment of semantic control has been more common in fMRI and neuropsychological studies, rather than studies of individual differences. One of the most common methods to probe semantic control is to use feature-matching tasks [16,21–26]. Specifically, in feature-matching tasks, participants are required to match concepts based on specific shared properties. For example, a participant might be asked whether a *bone* or a *tongue* is the same colour as a *tooth*. In this case, both response options are semantically associated with the probe and they compete with one another for selection. Top-down semantic control processes are thought to resolve the competition, based on the requirements of the current task [27,28]. Semantic control demands in this task can be manipulated by varying the relative strength of semantic association between the probe and its target and distractor. For example, deciding whether *salt* has the same colour as *sugar* or *tree* requires little control because the strong semantic association between *salt* and *sugar* supports the decision and there is little competition from the distractor. However, determining whether *salt* has the same colour as *pepper* or *cloud* places high demands on control processes due to the strong but irrelevant *salt–pepper* relationship. More generally, the manipulation of associative strength between concepts is commonly used to test semantic control ability [16,21]. The detection of weak semantic associations (e.g. *candle–halo*) is thought to require goal-directed controlled search through the semantic store for the relevant information [27]. Feature-matching tasks do not probe the size of the semantic store since the concepts used are relatively high

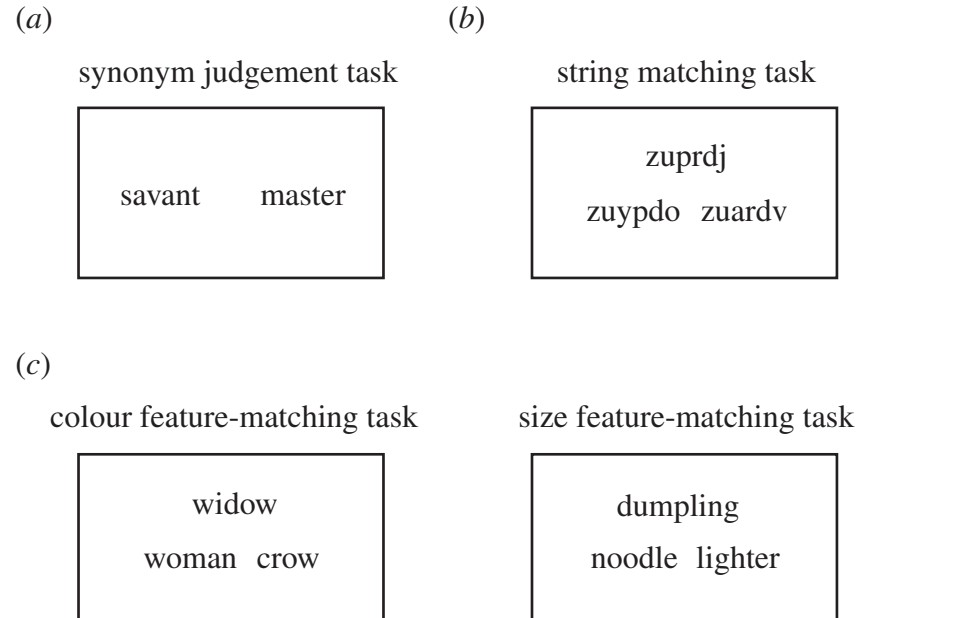

**Figure 1.** Example items from different tasks in the experiment. (*a*) Test of semantic knowledge, (*b*) test of non-semantic cognitive control and (*c*) test of semantic control.

in frequency and are assumed to be present within all participants' semantic knowledge. Performance instead depends on participants' ability to control activation within the semantic system such that the correct knowledge is accessed and used to drive the behavioural response.

To summarize, previous studies have adopted different experimental paradigms to test semantic knowledge and semantic control. However, unlike for semantic knowledge, there are no standardized assessments of semantic control ability that provide normative data for a specific set of stimuli. While many studies have used feature-matching tasks to test semantic control, each has adopted different stimuli and trial designs. The present study aims to address this by providing normative data for a large set of stimuli which tax both the knowledge and the control components of semantic cognition (e.g. figure 1). We devised a synonym judgement task to measure semantic knowledge representation (similar to the Mill Hill vocabulary scale and other vocabulary measures), which consisted of 160 trials. To minimize the demand on semantic control introduced by competition between multiple choices, we only included two words on each trial and participants were required to decide if the two words share a similar meaning or not. As a measure of semantic control ability, we devised a 160-item feature-matching task, since this task is commonly used in investigations of semantic control in healthy people and its control demands are well understood and quantifiable. To avoid feature-specific effects, we chose one of two properties (i.e. colour and size) as the target property in each half of the items in the feature-matching task. That is, the semantic control test was split into one 80-item colour feature-matching task and one 80-item size feature-matching task. Finally, for the researchers who are interested in the relationship between semantic control and non-semantic general cognitive control [3,21,26,29], the present norms also include a set of 160 string-matching task items. In the string-matching task, participants were asked to select the target string out of two non-word options based on their orthographic similarities with the probe string. This task requires participants to solve competition between targets and distractors without engaging semantic cognition. It thus works as a non-semantic comparison for the test probing semantic control ability.

We have collected behavioural responses (i.e. accuracy and reaction time (RT)) from more than 200 healthy older and young adult participants. Trial-level data for individual participants, as well as summary data for each of the 480 trials in the stimulus set are available on our OSF project page: https://osf.io/9px7g/. In this paper, we describe these data and a series of validation analyses performed on them. First, we report mean psycholinguistic properties of items in the tasks (i.e. frequency, target versus distractor strength (TDS), concreteness and word length) as well as mean performance of participants. Considering the potential utilization of our norms in ageing studies, we report the overall performance of each age group on each test separately and compare the

performance of different age groups in different tests using mixed effects models. Second, we demonstrate that our data replicate well-established effects of trial properties on knowledge-based and control-based semantic tasks. Previous research indicates that for knowledge-based tasks, performance is poorer for lower frequency words, while for control-based tasks, the degree of competition between target and distractor items is critical [21,23,25,30]. We demonstrate that both of these effects are present in our test items. We also replicate recent findings indicating that the effects of these variables interact with age [16,17]. Finally, we provide a subset of 80 items for each test grouped into five levels of difficulty. This parametric manipulation of difficulty will aid selection of materials for researchers who are interested in using our tests to study semantic processing in healthy or impaired populations. It also provides researchers with validated difficulty-graded materials to investigate age-related semantic ability changes. Rigorous parametric control of task demands is particularly important in studies of cognitive ageing, since neurocognitive models make complex and sometimes nonlinear predictions about their effects in different age groups (e.g. the CRUNCH model) [31].

# 2. Material and methods

## 2.1. Participants

One hundred and twenty-seven young adults (104 females; mean age = 19.29 years, s.d. = 3.80 years, range = 18–42) were recruited from the undergraduate Psychology course at the University of Edinburgh and participated in the study in exchange for course credit. One hundred and six older adults (77 females; mean age = 68.49 years, s.d. = 5.76 years, range = 60–85) were recruited from the Psychology department's volunteer panel and participated for payment. The levels of formal education were high in both groups (young adults: mean = 13.31 years, s.d. = 1.12 years, range = 11–17; older adults: mean = 15.46 years, s.d. = 2.65 years, range = 11–21), and older adults had completed more years of education than young adults ($t_{136.34} = 7.77$, two-tailed $p < 10^{-11}$). However, young adults were still participating in full-time education. All participants were native speakers of English and reported to be in good health with no history of neurological or psychiatric illness. Informed consent was obtained from all participants and the research was performed in accordance with all relevant guidelines/regulations. The study was approved by the University of Edinburgh Psychology Research Ethics Committee.

## 2.2. Materials

Three tests were completed by the participants, including a 160-item semantic knowledge test, a 160-item semantic control test and a 160-item non-semantic cognitive control test (figure 1 for examples; full set is provided on our OSF project page: https://osf.io/9px7g/).

### 2.2.1. Test of breadth of semantic knowledge

Participants completed a 160-item synonym judgement task designed to probe the size of their store of semantic knowledge. Specifically, on each trial, participants were presented with a pair of words and asked to decide if the two words shared a similar meaning or not. This task was designed to vary the frequency of the words and included some unusual words that were expected to be unknown to some members of the population. It therefore indexed the breadth of semantic knowledge available to each individual. Some stimuli for this synonym judgement task were identified from previous studies [7,16,17,23] and were modified to align with the current experimental paradigm. These were supplemented with additional items we generated specifically for this study. Log word frequencies for all words were obtained from the SUBTLEX-UK database [32], which produced two frequency values for each pair of words. Here, we chose the lower frequency in each word pair as a measure of the word frequency for this task item. We ensured the final stimuli included a wide range of word frequencies, in order to vary the demands placed on the semantic knowledge system. Thus we assumed that comprehension of lower frequency words place greater demands on this system [33]. By contrast, it is not the case that lower frequency words place greater demands on semantic control processes. In fact, some previous studies have indicated that *higher* frequency words place higher demands on control [34,35].

### 2.2.2. Test of semantic control

The semantic control ability of participants was probed with two 80-item feature-matching tasks, using a similar paradigm to prior studies [16,21,25]. Specifically, on each trial of the tasks, participants were presented with a probe word and asked to select the option that matched on particular features with the probe (either colour or size) from two alternatives. For example, on a colour trial, *salt* would be matched with *dove* as both are typically white. These tasks place high demands on semantic control ability because automatic activation of semantic knowledge is often not sufficient to identify the correct option: participants must direct their attention to the target semantic properties and ignore other semantic associations. The need for control can be manipulated by varying the relative strengths of the probe's semantic relationship with target and distractor. When the probe has a strong relationship with the target (e.g. is *celery* the same colour as *lettuce* or *dishwasher*?), little control is needed as the existing semantic associations support selection of the correct response. Conversely when the probe is semantically related to the distractor but not the target (e.g. is *salt* the same colour as *cloud* or *pepper*?), control demands are high because the strong pre-existing semantic association must be inhibited. To quantify the strength of the competition between target and distractor words in each of the items, the TDS was calculated following the method in a previous study [17]. In brief, the quantitative measures of the strength of the semantic relationships between probe and target/distractor words were obtained from a publicly available set of word representations generated by the word2vec neural network [36], by computing the cosine similarities of the word2vec vectors of the corresponding words. Then we calculated TDS for each task item as the strength of the probe–target relationship minus the probe–distractor relationship. We ensured our stimuli covered a wide range of TDS.

In addition to word frequency and TDS, word-average concreteness and sum length of words were also calculated for each item in semantic tests. Psycholinguistic properties of items in the two semantic tests were compared with each other. The items used in the semantic knowledge test were significantly lower in frequency ($t_{229.14} = -6.36$, two-tailed $p < 10^{-8}$), concreteness ($t_{163.90} = -13.83$, two-tailed $p < 10^{-28}$), and summed word length ($t_{318} = -10.81$, two-tailed $p < 10^{-22}$) than those used in the semantic control test.

### 2.2.3. Test of non-semantic cognitive control

As a comparison for the test probing semantic control ability, a 160-item string-matching task was completed by the participants as well, which examined general executive control ability during orthographic processing. The stimuli in this task were 160 triads that consisted of three letter strings. Participants were required to choose the option that shared the most letters in the same order as the probe. The stimuli for this string-matching task were generated with a Matlab script, following a set of rules which manipulated the letters presented in the triads and the orders of the letters (table 1). These varied stimuli such that the targets became increasingly dissimilar to the probes while the distractors became more similar, thus manipulating competition between options in a similar fashion to the semantic control test.

## 2.3. Procedure

Participants were tested via an online testing platform (https://www.testable.org/). The study could only be completed on a PC or laptop and not on a mobile device and participants were instructed to respond with their keyboard. The experiment consisted of four blocks of trials, one for each of the four tasks (synonym judgement, colour matching, size matching and string-matching tasks). The order of the blocks as well as the order of the trials shown in each block was randomized for each participant. Each block of trials began with a set of instructions and examples of correct responses (see electronic supplementary material for the full text of the instructions). Each trial began with a fixation cross presented for 500 ms, followed by a word pair/triad. For the synonym judgement task, the paired words separately appeared on the left and right of the centre of the screen; for the feature-matching and string-matching tasks, the probe word appeared above the centre of the screen with the two option words in a line below. Participants indicated their choice by button press (s or d for similar or dissimilar word pairs in the synonym judgement task; 1 or 2 for the options on the left or right in the feature-matching and string-matching tasks), and their response accuracy and RT were recorded. The position of the target (left, right) was balanced for the matching tasks and the number of similar and dissimilar word pairs was balanced for the synonym task. Participants were instructed

**Table 1.** Spelling rules for stimuli in the string-matching task. Note: there are 32 items under each spelling rule. (T) indicates the target in the item.

| rule | description | example | |
|---|---|---|---|
| 1 | Probe has five letters; target has five letters, four of the letters are from the probe (in the same order as the probe); distractor has five letters that are different from the probe. | xgbfj<br>xgqfj(T) | oucnz |
| 2 | Probe has five letters; target has five letters, four of the letters are from the probe (in the same order as the probe); distractor has five letters, two of the letters are from the probe (in the same order as the probe). | gxrcm<br>gvreo | gxrym(T) |
| 3 | Probe has five letters; target has five letters, three of the letters are from the probe (in the same order as the probe); distractor has five letters, two of the letters are from the probe (in the same order as the probe). | qlezj<br>qymzj(T) | qbesp |
| 4 | Probe has five letters; target has five letters, three of the letters are from the probe (in the same order as the probe); distractor has five letters, two of the letters are from the probe (in the same order as the probe), one of the letters is from the probe but locates at a different position. | ivazl<br>ipavb | ivuzm(T) |
| 5 | Probe has six letters; target has six letters, four of the letters are from the probe (in the same order as the probe); distractor has six letters, three of the letters are from the probe (in the same order as the probe), one of the letters is from the probe but locates at a different position. | btnfuk<br>btnyur(T) | btnuic |

to respond as quickly as possible without making mistakes. They were encouraged to guess if unsure of the correct response. No time limit was placed on responses.

## 2.4. Statistical analyses

There were three sections of analyses in the current study, based on accuracy and RT data of the participants. Before analysing the data, RTs of all trials were first screened for outliers for each participant and each task, by excluding the trials that took a participant an extremely long time to respond (longer than 30 s) and winsorizing any RTs more than two standard deviations from a participant's mean RT in the same task. The winsorized RTs together with accuracy rates of participants were combined to define the difficulty levels of items in the third section of analyses. It is worth noting that we also screened the raw RTs (i.e. excluding extremely long-time trials and winsorizing the RTs) for correct trials only (4.9% of the trials were screened), and RTs screened in this manner were then log-transformed and used as input for all RT-based linear mixed effects models. In other words, only RTs of correct responses were analysed in the mixed effects models. The RTs screened in the second manner were also used as input for descriptive statistical analyses (e.g. calculating mean RT).

In the first set of analyses, we aimed to describe the general attributes of our stimuli. First, we calculated overall performance (i.e. accuracy and RT) for each age group and each test separately. That is, we averaged the performance of all participants in the same group on each item and then averaged across items in each task. Second, overall effects of age group and test were assessed using mixed effects models. These models used age group and test type as predictors to predict participants' responses. To examine the interaction effects of the predictors, *post hoc* tests were further performed with additional mixed effects models, which included age group as a predictor to predict participants' performance in each test. In the last part of this section, we explored the relationships between performance on the different tests. Here, we computed Pearson's correlations with participants' performance on different tests within each age group separately. This analysis was performed on accuracy data only, as RTs were likely to covary due to variations in general processing speed that were not specific to a particular cognitive domain.

The second set of analyses examined effects of age group and specific psycholinguistic properties (as well as their interaction) on participants' performance, to test whether our stimuli replicated patterns found in previous studies of semantic cognition. In this set of analyses, separate mixed effects models

were estimated for the performance in semantic knowledge and semantic control tests. Specifically, for the semantic knowledge test (i.e. synonym judgement task), we used mixed effects models to predict accuracy and RT at the level of individual trials with age group as a between-subjects predictor and word frequency as a within-subject predictor. We included item type (i.e. similar- and dissimilar-word-pair items) as another within-subject predictor. To investigate performance on the semantic control test (i.e. colour and size feature-matching tasks), we used mixed effects models with age group and TDS as predictors to predict accuracy and RT in the two feature-matching tasks together.

In the final set of analyses, we aimed to generate a subset of items in each task that manipulated *difficulty levels* on a parametric basis.

We began by calculating a difficulty score for each test item, based on mean accuracy and RT. In each age group, behavioural performance on each item was averaged across the participants (separately for accuracy and RT). Next, we excluded items with low group-average accuracy (less than 60%) or long group-average RT (greater than 7 s) in either age group from further analyses, since participants did not appear to respond consistently to these items. For the remaining items, the group-average accuracy and RT were Z-transformed within each task, and a difficulty score was obtained for each item by subtracting the Z-score of accuracy from the Z-score of RT on this item and dividing this difference by two. Thus, items that had relatively higher RTs and relatively lower accuracy rates received higher difficulty scores. Lastly, a final difficulty score was calculated for each individual item by averaging the corresponding item's difficulty scores in young and older groups. The difficulty score for each item can be expressed as follows:

$$D = \frac{1}{2}\left(\frac{Z_{RT}^{Old} - Z_{Acc}^{Old}}{2} + \frac{Z_{RT}^{Young} - Z_{Acc}^{Young}}{2}\right). \tag{2.1}$$

Having computed an overall difficulty score for each item, we evenly sorted the items in each of the tasks into five difficulty levels and we selected the top 16 items (or top 8 for the two feature-matching tasks) of each difficulty level in each task to generate five sets of items. For the synonym judgement task, the subset selection was performed for the similar and dissimilar items separately to ensure that each subset contained an equal number of each type. To verify that this process was successful in manipulating task difficulty in both age groups, we then tested mixed effects models using the difficulty level as a predictor of performance in young and older groups. Given that the difficulty level was defined for each task separately, in this section of analyses, we built separate models for the two semantic-control-based tasks (i.e. colour and size feature-matching tasks).

Mixed effects models were constructed and tested using the recommendations of Barr *et al.* [37]. Logistic models were estimated for analyses of accuracy, and linear models were specified for analyses of RT. We specified a maximal random effects structure for all models, including random intercepts for participants and items as well as random slopes for all predictors that varied within-participant or within-item. The age group, test type and item type were included in the relevant models as categorical predictors, and the psycholinguistic property and difficulty level were included as continuous predictors. We also included the trial order in experiment in each model as a covariate of no interest. Continuous predictors were standardized prior to entry in the model. For the accuracy models, the statistical significance of effects of interest was assessed by comparing the full model with a reduced model that was identical in every respect except for the exclusion of the effect of interest. Likelihood-ratio tests were used to determine whether the inclusion of the effect of interest significantly improved the fit of the model. For the RT models, the statistical significance of effects of interest was assessed with the lmerTest package [38] via Satterthwaite's method.

# 3. Results

## 3.1. General performance of participants

Table 2 shows the psycholinguistic characteristics of the items used in each test, as well as mean performance in each age group. The s.d. and range in table 2 are calculated over items rather than participants. It therefore indicates that, while mean accuracy was high, there were a small number of items in each task for which accuracy was rather low. Items with mean accuracy less than 60% were removed when generating difficulty-graded subsets of the stimuli but are included in other analyses. We performed split-half correlation analyses to assess the reliability of the trial-level performance

**Table 2.** Mean psycholinguistic properties of task items and participants' performance. Note: word frequencies (on the log-transformed Zipf scale) were obtained from the SUBTLEX-UK database [32], and the lowest frequency in each word pair/triad was used as a measure of word frequency for this task item. Concreteness values were obtained from Brysbaert et al. [39] and the word-average concreteness was used as a measure of concreteness for each task item. TDS values were calculated in a similar fashion to a previous study [17]. Note that for the synonym task, the TDS was calculated for similar- (top row) and dissimilar-word-pair (bottom row) items separately, as the two words' word2vec distance. Word length was measured for each item by summing the number of characters over all words in the item. Mean RTs were group-average winsorized reaction times on all correct trials (except extremely long-time trials). Mean accuracy and RT were calculated by averaging participants' performance for each item and then averaging performance across items in each task. All s.d. and ranges were calculated over items instead of participants.

| | semantic knowledge test (synonym judgement task) | | | semantic control test (feature-matching tasks) | | | non-semantic cognitive control test (string-matching task) | | |
|---|---|---|---|---|---|---|---|---|---|
| | mean | s.d. | range | mean | s.d. | range | mean | s.d. | range |
| frequency (Zipf) | 2.79 | 1.13 | 0.70–5.44 | 3.42 | 0.54 | 2.09–4.98 | | | |
| TDS | 0.43 (similar) | 0.15 | 0.13–0.76 | 0.02 | 0.36 | −0.79–0.71 | | | |
| | 0.12 (dissimilar) | 0.10 | −0.05–0.45 | | | | | | |
| concreteness | 3.61 | 1.03 | 1.52–5 | 4.76 | 0.17 | 3.92–5 | | | |
| word length | 13.19 | 3.33 | 6–23 | 17.05 | 3.05 | 11–23 | 15.60 | 1.20 | 15–18 |
| young accuracy | 0.80 | 0.22 | 0.16–1 | 0.87 | 0.12 | 0.33–1 | 0.83 | 0.17 | 0.31–1 |
| young RT (ms) | 1957 | 638 | 1000–4174 | 2951 | 456 | 1867–4064 | 3104 | 1024 | 1618–6038 |
| older accuracy | 0.90 | 0.16 | 0.13–1 | 0.90 | 0.14 | 0.13–1 | 0.90 | 0.11 | 0.51–1 |
| older RT (ms) | 2030 | 535 | 1215–3896 | 3656 | 814 | 2029–6487 | 6191 | 2046 | 3082–10406 |

**Table 3.** Correlations of task performance (accuracy rates) in older and young groups. Note: *p*-values are uncorrected.

| | older | | young | |
|---|---|---|---|---|
| | *r*-value | *p*-value | *r*-value | *p*-value |
| semantic knowledge test – semantic control test | 0.19 | 0.05 | 0.33 | $<10^{-3}$ |
| semantic knowledge test – non-semantic cognitive control test | 0.11 | 0.26 | 0.37 | $<10^{-4}$ |
| semantic control test – non-semantic cognitive control test | 0.50 | $<10^{-7}$ | 0.29 | $<10^{-3}$ |

estimates in each age group and each test. Specifically, for each test, we repeatedly divided young or older subjects into two groups, calculated mean performance (i.e. RTs or accuracy) on each test item in each group, and then computing the correlation between these two halves of the data (5000 splits). We then averaged these split-half correlations over iterations. This analysis revealed that, the behavioural responses were extremely reliable in both age groups and in each test: $rs > 0.81$.

To compare the performance of the two age groups in different tests, mixed effects models with age group and test type as predictors were constructed and tested. We found that, in general, older people produced significantly more accurate but slower responses than young people (accuracy: $B = 0.401$, s.e. $= 0.045$, $p < 10^{-15}$; RT: $B = 0.086$, s.e. $= 0.008$, $p < 10^{-15}$). There was also a significant test type effect on RT ($p < 10^{-15}$) and interaction effects on both accuracy and RT (accuracy: $p < 0.01$; RT: $p < 10^{-15}$). To investigate the interaction effects, mixed effects models with age group as a predictor of performance were further computed for each test as *post hoc* analyses. The results revealed that, although older people were significantly more accurate and slower in all the tests, the accuracy difference between young and older participants was smaller in the semantic control test than that in the other tests (knowledge test: $B = 0.485$, s.e. $= 0.052$, $\chi^2 = 73.10$, $p < 10^{-15}$; semantic control test: $B = 0.287$, s.e. $= 0.042$, $\chi^2 = 47.85$, $p < 10^{-11}$; non-semantic control test: $B = 0.349$, s.e. $= 0.029$, $\chi^2 = 82.03$, $p < 10^{-15}$). The biggest group difference on RT was in the non-semantic cognitive control test and the smallest was in the semantic knowledge test (knowledge test: $B = 0.029$, s.e. $= 0.002$, $t_{147.74} = 13.02$, $p < 10^{-15}$; semantic control test: $B = 0.051$, s.e. $= 0.002$, $t_{151.54} = 24.14$, $p < 10^{-15}$; non-semantic control test: $B = 0.169$, s.e. $= 0.002$, $t_{157} = 76.79$, $p < 10^{-15}$). These results largely confirm prior theories about age-related differences in semantic cognition. On the knowledge task, older people were much more accurate than young people and only slightly slower, suggesting they have access to a wider range of semantic knowledge. In the semantic control task, in contrast, older people were only slightly more accurate and were much slower than young people, suggesting that control over semantic processing is an area of relative weakness in old age.

We also explored the relationships among the tests in our norms. The correlations of test performance (accuracy rates) in each age group are presented in table 3. Within young participants, performance on the three tests was all moderately positively correlated. Interestingly, within older participants, although performance on the semantic control test and the non-semantic cognitive control test were significantly correlated, the two semantic tests showed a weak relationship with each other, indicating that the semantic knowledge and semantic control tests indexed different aspects of semantic cognition, especially for the older group. These results underscore the need to measure semantic knowledge and semantic control ability separately as these abilities are only weakly associated with one another.

## 3.2. Effects of age group and psycholinguistic properties on semantic cognition

In this section of analyses, we validated the reliability of our norms by replicating prior findings, which illustrated age- and psycholinguistic-property-related effects on different aspects of semantic cognition [16,17].

For the knowledge-based task, the accuracy and RT data were analysed in mixed models that included age group as a between-subjects predictor and frequency and item type (similar or dissimilar) as within-subject predictors. Note that since the full model for accuracy data here failed to converge, we used a simplified model which removed the correlations between random effects. The parameter estimates for the accuracy and RT models are presented in table 4. Consistent with previous results, older people were slower to respond but more accurate. We also found that lower frequency led to slower and less accurate performance in general, replicating well-established effects of word frequency on performance [16,17,30]. Interestingly, the results showed that dissimilar word

**Table 4.** Mixed effects models predicting participant accuracy and RT in the semantic knowledge test from age group, frequency and item type.

| effect | accuracy | | | RT | | |
|---|---|---|---|---|---|---|
| | B | s.e. | p | B | s.e. | p |
| group | 0.520 | 0.058 | $<10^{-15}$ | 0.031 | 0.009 | $<0.001$ |
| frequency | 0.867 | 0.112 | $<10^{-13}$ | $-0.061$ | 0.006 | $<10^{-15}$ |
| item type | 0.239 | 0.118 | $<0.05$ | 0.028 | 0.006 | $<10^{-5}$ |
| group * frequency | $-0.038$ | 0.054 | 0.618 | 0.003 | 0.003 | 0.315 |
| group * item type | 0.025 | 0.068 | 0.598 | $-0.006$ | 0.003 | 0.064 |
| frequency * item type | $-0.381$ | 0.113 | $<0.001$ | 0.019 | 0.005 | $<0.001$ |
| group * frequency * item type | 0.248 | 0.058 | $<10^{-5}$ | $-0.008$ | 0.002 | $<0.001$ |

pairs caused slower but more accurate responses than similar ones, and frequency had stronger influences on performance on similar word pairs than that on dissimilar word pairs (in both accuracy and RT). Importantly, there were highly significant three-way interactions between these predictors, suggesting that different item types had effects on the other two variables of interest.

The nature of the above interactions was investigated with separate mixed effects models (age group × frequency) for each item type. The modelled effects of the predictors are shown in figure 2. For the similar word pairs, older people performed more accurately ($B = 0.526$, s.e. $= 0.104$, $p < 10^{-6}$) but slower ($B = 0.037$, s.e. $= 0.008$, $p < 10^{-5}$) than young people. The effect of frequency was significant in both accuracy and RT models (accuracy: $B = 1.387$, s.e. $= 0.169$, $p < 10^{-13}$; RT: $B = -0.083$, s.e. $= 0.008$, $p < 10^{-15}$). Additionally, we found that the frequency effect was more pronounced in the young group (accuracy: $B = -0.271$, s.e. $= 0.093$, $p < 0.01$; RT: $B = 0.012$, s.e. $= 0.004$, $p < 0.01$), replicating previous findings [17]. For the dissimilar word pairs, the age group and frequency exhibited similar main effects on accuracy and RT as prior results (group effect on accuracy: $B = 0.553$, s.e. $= 0.082$, $p < 10^{-10}$; on RT: $B = 0.026$, s.e. $= 0.011$, $p < 0.05$. Frequency effect on accuracy: $B = 0.515$, s.e. $= 0.156$, $p < 0.001$; on RT: $B = -0.041$, s.e. $= 0.008$, $p < 10^{-6}$). However, the interaction effects showed a converse result pattern (significant in accuracy only) comparing with the analyses with only similar word pairs (accuracy: $B = 0.214$, s.e. $= 0.068$, $p < 0.001$; RT: $B = -0.005$, s.e. $= 0.003$, $p = 0.123$), that is, young people's performance was less affected by frequency in comparison with older people in dissimilar items.

For the semantic-control-based tasks, two mixed models that included age group as a between-subjects predictor and TDS as a within-subject predictor were estimated for the accuracy and RT data, respectively. The modelled effects of the predictors in the accuracy and RT models are shown in figure 3. Specifically, for accuracy, the main effects of age group and TDS as well as their interaction were all significant (group: $B = 0.279$, s.e. $= 0.072$, $p < 10^{-4}$; TDS: $B = 0.559$, s.e. $= 0.091$, $p < 10^{-9}$; interaction: $B = 0.133$, s.e. $= 0.053$, $p < 0.01$). A similar pattern was found for the RT model (group: $B = 0.053$, s.e. $= 0.008$, $p < 10^{-10}$; TDS: $B = -0.027$, s.e. $= 0.006$, $p < 10^{-5}$; interaction: $B = -0.006$, s.e. $= 0.002$, $p < 0.01$). These results suggest that, as expected, participants were slower and less accurate to respond to low TDS items, in which the distractors were more strongly related to the probes than the targets. In addition, older people were more strongly influenced by this factor than young people, replicating previous findings [16,17].

## 3.3. Relationships between difficulty level and behavioural performance

As described in the Material and methods, we used performance data to define a subset of 80 trials for each test, divided into five levels of difficulty (16 items per level). In this section of analyses, we first examined if these difficulty levels were a valid indicator of the performance of each age group in each task as expected. Then we performed descriptive statistical analyses to illustrate the general attributes of the stimulus subsets.

In the first analysis, for young people, their accuracy and RT data in different tasks were separately analysed in eight mixed models that included difficulty level as a within-subject predictor.

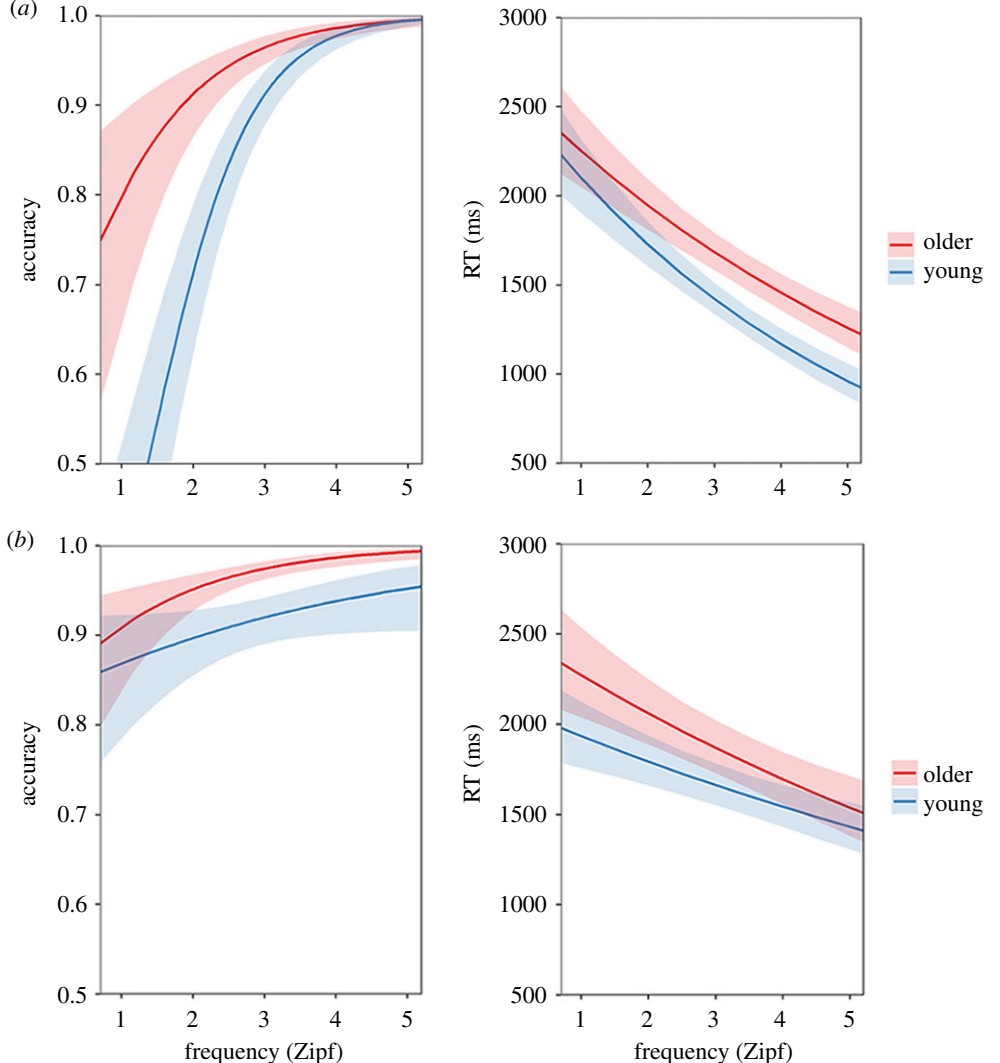

**Figure 2.** Modelled effects of age group and frequency on accuracy and RT for similar- (*a*) and dissimilar-word-pair (*b*) items in the semantic knowledge test. Shadow areas indicate 95% confidence intervals.

Parameter estimates for the accuracy and RT models are presented in table 5, and the modelled effects of the predictor are shown in figure 4. There were significant effects of difficulty level on all the models, that is, participants produced slower and less accurate responses as the difficulty level increased. For older people, since the full model for accuracy data in the colour task failed to converge, we used a simplified model to calculate the relevant results. Mixed effects models revealed similar result patterns to young group in the older people (table 5 and figure 4). These results indicate that the difficulty-graded subsets of items effectively manipulate task demands in both groups of participants.

We also replicated the above difficulty level effects with behavioural data from another group of subjects (number of subjects = 21, 17 females; mean age = 57.86 years, s.d. = 7.10 years, range = 50–75), whose responses were not used to construct our norms or define the difficulty levels. Parameter estimates for the mixed effects models and the modelled effects of difficulty level in the replication sample are shown in the electronic supplementary material. Significant effects of difficulty ($p < 10^{-6}$) were observed for accuracy and RT in all tasks.

Table 6 shows mean performance data and example trials for each subset of stimuli. Additionally, we plotted histograms showing the distribution of accuracy in each age group for each task and each difficulty level separately (figure 5). This illustrates that the tasks are sensitive to individual differences between participants, particularly at the higher difficulty levels where ceiling effects are avoided.

Lastly, correlations between difficulty levels and the psycholinguistic properties were calculated for the items in the semantic tests. We found that the difficulty levels in the knowledge test were significantly correlated with frequency ($r = -0.56$, $p < 10^{-7}$) but not with semantic distance (i.e.

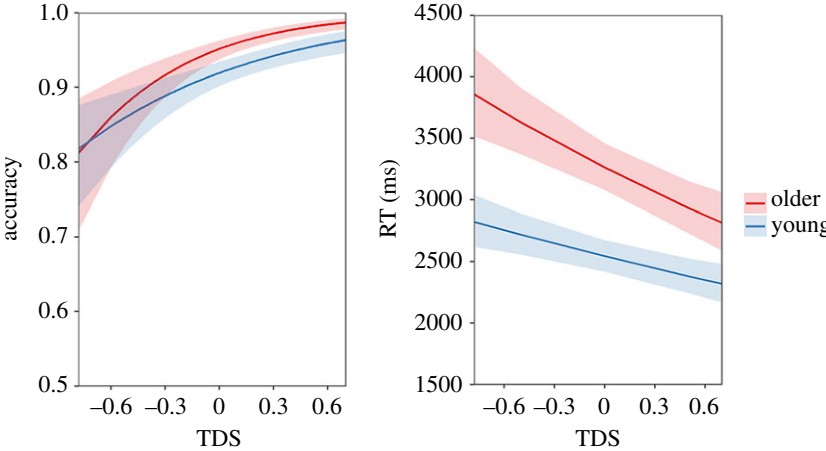

**Figure 3.** Modelled effects of age group and TDS on accuracy and RT in the semantic control test. Shadow areas indicate 95% confidence intervals.

**Table 5.** Mixed effects models predicting participant accuracy and RT from difficulty level.

| | difficulty effect on accuracy | | | difficulty effect on RT | | |
|---|---|---|---|---|---|---|
| | B | s.e. | p | B | s.e. | p |
| *older group* | | | | | | |
| colour feature-matching task | −0.983 | 0.081 | $<10^{-15}$ | 0.069 | 0.003 | $<10^{-15}$ |
| size feature-matching task | −1.394 | 0.096 | $<10^{-15}$ | 0.059 | 0.003 | $<10^{-15}$ |
| synonym judgement task | −1.401 | 0.084 | $<10^{-15}$ | 0.064 | 0.003 | $<10^{-15}$ |
| string-matching task | −1.139 | 0.123 | $<10^{-14}$ | 0.086 | 0.003 | $<10^{-15}$ |
| *young group* | | | | | | |
| colour feature-matching task | −0.941 | 0.069 | $<10^{-15}$ | 0.055 | 0.003 | $<10^{-15}$ |
| size feature-matching task | −0.938 | 0.065 | $<10^{-15}$ | 0.040 | 0.003 | $<10^{-15}$ |
| synonym judgement task | −1.117 | 0.053 | $<10^{-15}$ | 0.058 | 0.003 | $<10^{-15}$ |
| string-matching task | −1.138 | 0.066 | $<10^{-15}$ | 0.069 | 0.003 | $<10^{-15}$ |

word2vec) of the paired words ($r = -0.01$, $p = 0.96$). Note that we also calculated the correlation between difficulty levels and semantic distance for each type of word pairs separately, the correlations were not significant either (similar word pairs: $r = -0.16$, $p = 0.32$; dissimilar word pairs: $r = 0.10$, $p = 0.55$). By contrast, the difficulty levels in the semantic control test were correlated with TDS ($r = -0.26$, $p < 0.05$) but not frequency ($r = 0.06$, $p = 0.61$). This suggests that the above psycholinguistic properties can be successfully used as indicators of task demands in corresponding semantic tests, which validated our strategy regarding stimulus selection.

## 4. Discussion

Effective semantic cognition relies on the store of semantic knowledge as well as on semantic control processes that regulate retrieval and manipulation of this knowledge. Although there are many standardized assessments of semantic knowledge, standardized tests that measure semantic control ability do not exist. In this study, we described a normed stimulus set with tasks that probe both the knowledge and the control components of semantics. Behavioural performance of over 200 young and older participants on our stimulus set were collected. Using these data, we replicated a series of established psycholinguistic property effects on the semantic knowledge and semantic control tasks in

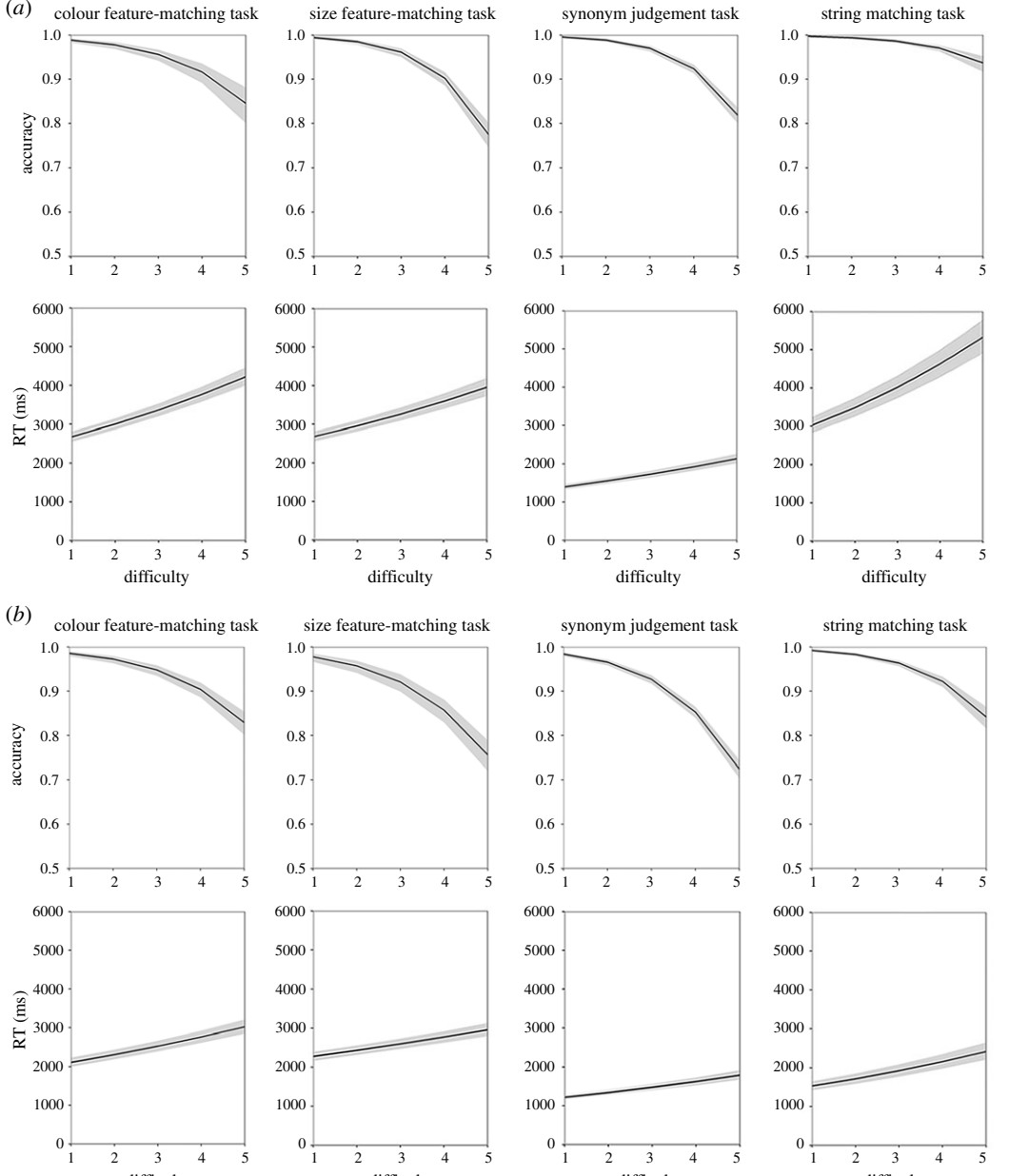

**Figure 4.** Relationships between difficulty level and behavioural performance of older (*a*) and young (*b*) participants on subset stimuli in different tasks. Shadow areas indicate 95% confidence intervals.

different age groups, which validated the reliability of our norms. We also used the behavioural data to define parametrically varied difficulty levels for a subset of the stimuli, which will allow future studies to investigate semantic processing at different levels of demand.

The purpose of our norms is to provide researchers in the fields of cognitive neuroscience and cognitive ageing with a resource they can use to systematically select stimuli to test semantic abilities. The availability of a normed resource like our stimulus set has the advantage of providing a more standardized approach for stimulus selection, and if adopted more widely, offers an increased level of comparability between different studies. There are a wide range of potential applications of our norms. First, our results have shown that our stimuli have a sensitivity to individual variation in semantic abilities, especially as the task difficulty increases. Thus, our difficulty-graded tests can be used for assessment of semantic knowledge and control in healthy populations. Depending on time available for assessment and the purpose of the study, researchers can flexibly use our stimuli at different difficulty levels to generate customized assessments to test semantics. For example, studies that focus primarily on accuracy may wish to exclude the easier levels of difficulty, which are more

**Table 6.** Test performance of participants on subset items at each difficulty level in each task. Note: mean RTs were group-average winsorized reaction times on all correct trials (except extremely long-time trials). Mean accuracy and RT were calculated by averaging participants' performance for each item and then averaging performance across items at each difficulty level in each task. All s.d.s were calculated over items instead of participants. (T) indicates the target in the item. (S) and (D) indicate similar-word-pair and dissimilar-word-pair items.

| | | young accuracy | | young RT (ms) | | older accuracy | | older RT (ms) | |
|---|---|---|---|---|---|---|---|---|---|
| difficulty | example | mean | s.d. | mean | s.d. | mean | s.d. | mean | s.d. |
| semantic knowledge test (synonym judgement task) | | | | | | | | | |
| 1 | dliche rug (D) / prawn shrimp (S) | 0.97 | 0.03 | 1371 | 243 | 0.99 | 0.01 | 1514 | 135 |
| 2 | hog moped (D) / fastener zipper (S) | 0.96 | 0.02 | 1550 | 246 | 0.98 | 0.01 | 1654 | 208 |
| 3 | nemesis president (D) / aspirin tablet (S) | 0.93 | 0.04 | 1754 | 318 | 0.97 | 0.03 | 1926 | 191 |
| 4 | gonorrhea headache (D) / tendency trend (S) | 0.84 | 0.08 | 1886 | 432 | 0.93 | 0.07 | 2018 | 224 |
| 5 | felicitous faithful (D) / denture implant (S) | 0.72 | 0.07 | 2392 | 416 | 0.81 | 0.10 | 2530 | 519 |
| semantic control test (colour feature-matching task) | | | | | | | | | |
| 1 | walnut / breeze almond(T) | 0.98 | 0.01 | 2412 | 115 | 0.98 | 0.03 | 2971 | 281 |
| 2 | lemon / highlighter(T) tree | 0.94 | 0.03 | 2561 | 246 | 0.94 | 0.03 | 3272 | 464 |
| 3 | lead / pencil graphite(T) | 0.92 | 0.02 | 3122 | 261 | 0.95 | 0.04 | 3772 | 235 |
| 4 | moon / earth tooth(T) | 0.91 | 0.03 | 3098 | 211 | 0.90 | 0.05 | 4034 | 611 |
| 5 | artery / cranberry(T) vein | 0.81 | 0.06 | 3548 | 350 | 0.79 | 0.07 | 4828 | 291 |

(Continued.)

**Table 6.** (*Continued.*)

| | difficulty | example | young accuracy | | young RT (ms) | | older accuracy | | older RT (ms) | |
|---|---|---|---|---|---|---|---|---|---|---|
| | | | mean | s.d. | mean | s.d. | mean | s.d. | mean | s.d. |
| semantic control test (size feature-matching task) | 1 | nurse hospital baboon(T) | 0.94 | 0.04 | 2580 | 157 | 0.99 | 0.01 | 2946 | 252 |
| | 2 | basketball head(T) player | 0.88 | 0.04 | 2817 | 132 | 0.97 | 0.01 | 3249 | 168 |
| | 3 | roof canopy(T) stool | 0.89 | 0.04 | 3141 | 257 | 0.96 | 0.04 | 3650 | 208 |
| | 4 | boot pigeon(T) car | 0.84 | 0.06 | 2966 | 219 | 0.89 | 0.04 | 3648 | 307 |
| | 5 | candle lantern cup(T) | 0.73 | 0.06 | 3555 | 267 | 0.77 | 0.05 | 4716 | 341 |
| non-semantic cognitive control test (string-matching task) | 1 | ksije ksvje(T) tbdnx | 0.99 | 0.01 | 1833 | 85 | 0.99 | 0.01 | 3531 | 265 |
| | 2 | ibavf imjuf ibayf(T) | 0.98 | 0.01 | 2028 | 116 | 0.99 | 0.01 | 3833 | 208 |
| | 3 | xpstj xhqtn xpstk(T) | 0.95 | 0.02 | 2329 | 132 | 0.99 | 0.01 | 4236 | 384 |
| | 4 | fyvjq fyejq(T) fyswx | 0.92 | 0.05 | 2669 | 172 | 0.98 | 0.02 | 5240 | 519 |
| | 5 | xlmueg xlcaem xjmued(T) | 0.82 | 0.09 | 3250 | 373 | 0.90 | 0.04 | 6678 | 311 |

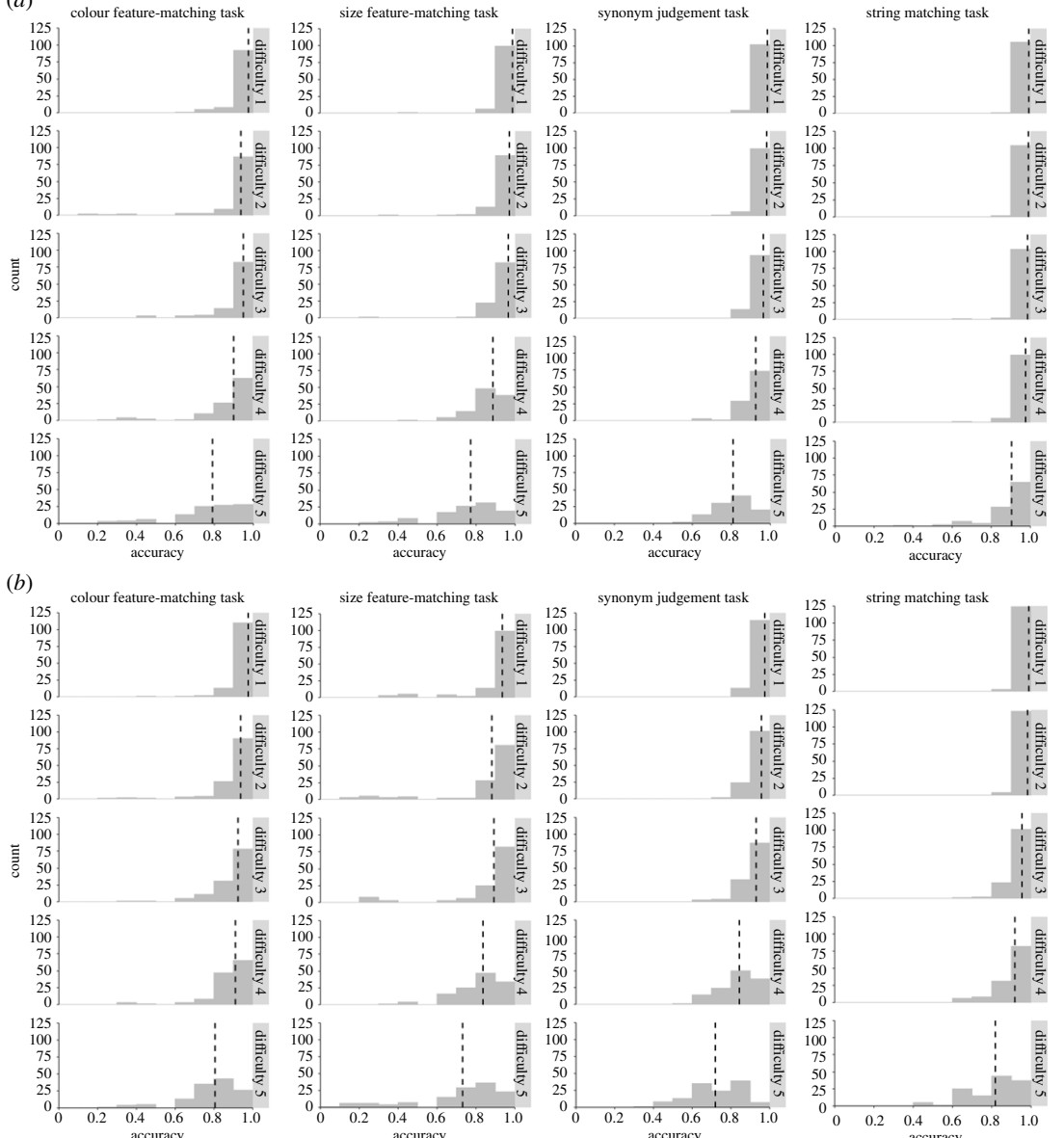

**Figure 5.** Distribution of older (*a*) and young (*b*) participants' overall accuracy rates on subset items in different tasks and at different difficulty levels. The height of the bars indicates number count of participants whose performance fell within a specified range of accuracy values. Dash line indicates group-average accuracy.

prone to ceiling effects. Conversely, where RT is of interest, researchers may wish to focus on the easier levels which will yield the highest numbers of correct responses. Second, researchers can use our full set of task items to produce their own bespoke sets of experimental stimuli with particular characteristics. For example, as our normative data include average accuracy rates and RTs for all items, researchers will have the opportunity to match items for accuracy and RT while varying other properties of interest in their own studies.

Using the current norms, we replicated several established psycholinguistic property effects on participants' semantic-based task performance, which validated the reliability of our norms. We found that older adults had a performance advantage over young people on the semantic knowledge test, which was significantly reduced on the semantic control task. Within the semantic control test, we have identified TDS, a measure of the strength of the target–probe semantic relationship relative to irrelevant distractor–probe relationship, as a predictor that has greater influence on semantic processing in older adults compared with young adults. Older people's performance declined more dramatically when the target–probe semantic relationship was weak and competition from irrelevant distractor was strong. Identifying the correct response under the low TDS conditions places high

demands on semantic control processes [4,25,27], and similar results showing an old-age deficit in semantic control have been reported in previous studies [16–18].

In the semantic knowledge test, for the similar-word-pair items, we found a word frequency effect that was consistent with previous research [17]. That is, as word frequency in task items decreased, young people had more difficulty in processing the meanings of words, whereas frequency had less influence on the performance of older people. Interestingly, a converse pattern was found for the dissimilar-word-pair items, which showed that young people's task performance on these items was less affected by word frequency than older people's. This discrepancy could be related to strategies employed when one or more words on the trial were unfamiliar to participants. In these cases, we suspect that participants tended to automatically reject the word pair as 'dissimilar'. This response bias would reduce accuracy on similar trials, particularly for low-frequency words. This is precisely what we observed (figure 2). Because the young group had smaller and less-detailed repositories of semantic knowledge, as suggested by a number of previous studies [10–14], they presumably used this strategy much more often than their older counterparts, leading to a pronounced drop-off in accuracy for low-frequency similar trials. This response bias suggests that, in synonym judgement tasks, participants' responses could be more sensitive to semantic knowledge when there is a relationship present.

We used a feature-matching test to probe semantic control because this test is commonly used in investigations of semantic control in healthy individuals and the control demands in this test are well understood. However, the feature-matching test is not the only paradigm that has been used to manipulate control demands in semantic processing in previous studies. Another common method is to manipulate the context in which semantic decisions are made [40,41]. When a coherent contextual cue is provided prior to the decision, it allows the participant to activate relevant conceptual knowledge prior to the decision, reducing the requirement for top-down semantic control processes. On the contrary, cues that contain irrelevant information increase control demands by introducing conflicting conceptual information that has to be ignored. In other studies, researchers have tested semantic control ability by manipulating competition between multiple meanings of the same words (i.e. using homonyms). For example, Noonan and colleagues measured semantic control ability by comparing participants' performance on retrieving the less common meanings of ambiguous words with the more common meanings when participants were making semantic judgements (e.g. linking *bank* with *river* in comparison with linking *bank* with *money*), and the former situation was thought to need more control resources than the later one [41]; another study investigated the processing of high-ambiguity sentences (containing homonyms), since this processing was deemed as having additional requirements of selecting contextually appropriate word meanings in comparison with low-ambiguity sentences [42]. In neuropsychological studies, semantic control has also been investigated using 'refractory' effects in 'cyclical' semantic tasks [43–45]. In these tasks, sets of semantically related items are presented repeatedly at a fast rate, thus activation of meanings spreads between items and cannot fully decay between trials. As a result, the entire set of task items become highly active and this gives rise to strong competition between items, which leads to poorer processing in patients with semantic control deficits.

It is worth noting that, although all above tests can be used to probe semantic control ability, different tests may tap on different neural resources and/or different components of semantic control disproportionately. For example, we have suggested that feature-matching tasks may be largely served by a domain-general executive component of semantic control ability [16]. Indeed, our current results showed a correlation between participants' performance on the feature-matching and string-matching tasks, which suggested the engagement of the domain-general component of semantic control in our semantic control task. This argument is also aligned with the neuroimaging studies in which increased selection demands in semantic tasks were associated with activation in the posterior portion of left inferior prefrontal cortex (BA44/45) [21,25,46], an area also activated by conditions of high competition in working memory tasks [47], the Stroop task [48] and other inhibitory tasks [49]. On the contrary, some other tasks (e.g. controlled retrieval of weak semantic links) appear to be more related to a semantic-specific component of semantic control. In those tasks, increased control demands are associated with increased activation in the anterior, ventral portion of left inferior prefrontal cortex (BA47), an area that is selectively activated by semantic tasks [21,46,50,51]. Despite these potential differences, there have been few direct comparisons of behavioural performance on different control tasks in healthy people. Therefore, one important target for future studies is to explore the relationships between different semantic control tasks and their supporting mechanisms.

Finally, for researchers who are interested in using our norms, it is important to note that aspects of test administration might influence the actual performance of participants. First, we collected behavioural data by presenting each task as a single block of trials. If the items from different tasks are presented in an interleaved fashion, behavioural responses might be influenced, as the switching between different tasks would introduce additional executive demands. Second, items of varying difficulties were intermixed within each task block. Participants' performance could be different when the task items are blocked by difficulty, as this could encourage strategy use. This may be particularly important in the semantic control tasks, since participants may become aware of blocks where general semantic association is helpful/unhelpful. Third, we did not impose a time limit while collecting the behavioural data, which may explain the strong age effects in our string-matching task. Older people took twice as long as young people on average in this task. However, they were considerably more accurate, suggesting that different age groups adopted different trade-offs between speed and accuracy. Placing limits on the time to complete each trial would be expected to shift the speed–accuracy trade-off [52].

In conclusion, we hope making our norms publicly available may prove to be useful for general studies of semantic processing, neuropsychological assessment and cognitive ageing, thereby further advancing the communication between these fields of research.

Ethics. The study was approved by the University of Edinburgh Psychology Research Ethics Committee. The procedures used in this study adhere to the tenets of the Declaration of Helsinki.

Data accessibility. The data reported in the study and code are available on the OSF website (https://osf.io/9px7g/). The study was not preregistered.

Consent to participate. Informed consent was obtained from all individual participants included in the study.

Authors' contributions. W.W.: conceptualization, data curation, formal analysis, investigation, methodology, validation, visualization, writing—original draft, writing—review and editing; P.H.: conceptualization, funding acquisition, investigation, methodology, project administration, supervision, validation, writing—review and editing.

All authors gave final approval for publication and agreed to be held accountable for the work performed therein.

Competing interests. None of the authors has conflicts of interest to declare.

Funding. The research was supported by a BBSRC (grant no. BB/T004444/1).

Acknowledgements. We are grateful to Beth Jefferies for sharing experimental stimuli.

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
