## [Peer Review File · Royal Society Open Science]

Review History

RSOS-211056.R0 (Original submission)

Review form: Reviewer 1

Is the manuscript scientifically sound in its present form?

Yes

Are the interpretations and conclusions justified by the results?

Yes

Is the language acceptable?

Yes

Do you have any ethical concerns with this paper?

No

Have you any concerns about statistical analyses in this paper?

No

Recommendation?

Accept as is

Comments to the Author(s)

Thank you for the opportunity to review this study, which makes an important and very useful contribution to the field of semantic cognition, across populations. I am sure that many research groups will benefit from the use of the norms provided by this study! This study administered a semantic knowledge (synonym judgment task) and a semantic control (feature-matching task) to younger (n=127) and older (n=106) adults and replicated previous findings that older adults tend to have a deeper semantic store, yet are out-performed by their younger counterparts on tasks that require efficient control over these representations. It also produces a set of normed data and demonstrates the relationship between TDS and performance (i.e., lower performance for low-TDS items), and how this effect is amplified in older individuals. I only have a few very minor comments below.

I noticed that your young age range is 18-42, however, based on your mean and SD, it looks like very few participants will be over 25. How many participants are 'outliers' in terms of age? Do your results change/is their behaviour different to the rest of the younger group? (I know the age range is large for the older age group - but this is more standard in the literature. It would still be interesting to see whether age has an effect within this group.)

Could you possibly provide information about proportion of RT's winsorized? For example, are there any trials that consistently needed winsorizing across participants?

Given the older adults performance on the semantic control and cognitive control tasks is correlated, but not the semantic control semantic knowledge; what implications does this have for the processes engaged to perform the semantic control task (i.e., feature matching) you use in this study? (e.g., does this task engage semantic control or executive control?) Maybe I missed this, but did you compare your synonym judgment task to other standardized assessments of semantic knowledge?

Do you have any citations to support your notion that a time-limit might encourage older people to respond more quickly at the cost of accuracy?

Review form: Reviewer 2

Is the manuscript scientifically sound in its present form?

No

Are the interpretations and conclusions justified by the results?

No

Is the language acceptable?

Yes

Do you have any ethical concerns with this paper?

No

Have you any concerns about statistical analyses in this paper?

Yes

Recommendation?

Reject

Comments to the Author(s)

This manuscript presents norms from 127 young and 106 older participants for two semantic tasks. The authors sustain that these tasks assess the organization of knowledge in the semantic store (synonym judgement task) and control processes involved in the retrieval of this knowledge (feature-matching task). They also affirm that their results are valid because they corroborate previous age- and linguistic-related effects.

In my view the main problem of this study is that it is described as a normative study. I find the experimental methodology and the analytical approach not solid enough for the aim of the study (i.e., building norms from young and older participants for semantic knowledge and control tasks). Moreover, I'm not sure that the tasks actually assess what the authors aim to assess. For these reasons, I will focus my review on the major critical points of the study. I am sorry I cannot be more positive but I hope that my comments can be helpful and constructive.

1) The authors write that the tasks dissociate the level of semantic knowledge and control processes, but I don't think this is the case. Indeed, the fact that the authors did not measure the relation between cue and target (in terms of association strength, lexical co-occurrence for example) in the synonym judgement doesn't mean that this relationship does not have an effect on participants' performance and that the retrieval of these links do not require control. It is reasonable to think that some word pairs are more associated (or more similar in meaning) compared to others, so for these latter, the controlled retrieval is needed to access to them. At some point in the Introduction the authors themselves write that the semantic control requires "flexible retrieval of less salient aspects of knowledge" (i.e., controlled retrieval) (pag. 5). For this reason, I would recommend the authors to measure the link between word pairs in the semantic judgments and avoid the division between semantic knowledge and control processes. Perhaps, it would be preferable to talk about semantic processing in general given the nature of the tasks.

2) Related to my previous point, in many tasks (like the synonym judgment task) requiring the processing of semantic information is really difficult to understand if the effects found are due to the manipulation of information at the representation level or manipulation of information for its retrieval. For example, how can the authors exclude that the word frequency effect found in the synonym judgement is due to the greater control demands required for the retrieval of low frequent words?

3) The tasks included in this study include only a small number of stimuli. To function as a norm, this study should have included a very large number of stimuli allowing researchers to select and manipulate word variables of interest (see for example, the Semantic Priming Project or other norms with linguistic stimuli, like Small World of Words for English language, and so forth).

4) Another issue I see is that the tasks should have been controlled for their general difficulty, especially because they assess the capacity of cognitive control demands (specific to the semantic domain and more general). This is also evident when we consider the semantic tasks only. Indeed, the authors write that "The items used in the semantic knowledge test were significantly lower in frequency [...], concreteness [...], and summed word length [...] than those used in the semantic control test." (pag. 9). More importantly, this difference among tasks emerged in the results (pag. 13, see main effect of test on RT and from the Table 6). I think this is a great issue in this study, especially given the dispute in literature about how much executive functions and semantic control overlap in terms of cognitive processes and brain underpinnings.

5) I wonder why the response mapping was different between the synonym judgement and the feature-matching task (pag. 10).

6) It is unclear to me how the authors built the linear mixed effect models. For example, were the confounding variables (at least the ones for which the semantic tasks differed) included in the analyses?

- 7) Pag. 14, line 14-22: this paragraph is more part of a Discussion section than a Results section.
- 8) There is no reliability measure of the data (i.e., split-half correlation). I would recommend the authors to perform these analyses to control if the data are consistent across participants and items.

Review form: Reviewer 3 (Hannah E. Thompson)

Is the manuscript scientifically sound in its present form?

Yes

Are the interpretations and conclusions justified by the results?

Yes

Is the language acceptable?

Yes

Do you have any ethical concerns with this paper?

No

Have you any concerns about statistical analyses in this paper?

No

Recommendation?

Accept with minor revision (please list in comments)

Comments to the Author(s)

This is a detailed battery, with analysis of a significant dataset, with openly available stimuli and data, warranting publication. The main element which could be further addressed is describing the stimuli in greater detail, particularly considering the task difficulty manipulation, as described below.

-The semantic knowledge task has a difficulty manipulation in terms of the frequency of words. The distinction with semantic control could be further addressed – e.g., are the words in the low frequency manipulation as related (in semantic space) as high frequency words? Semantic distance has previously been used as a difficulty manipulation (Noonan et al., 2010) and found to relate to semantic control deficits, so it would be worth including this detail. If semantic distance could be shown to be the same for high vs. low frequency words, the argument may be more compelling. It also seems that some literature regarding frequency and its relationship to control could be added (the reduced/ reversed frequency effect in SA - and papers on synonym judgement in SA).

-A further consideration to this end might be to consider co-occurrence (or frequency) of the two items combined (as previously explored, e.g., Jefferies & Lambon Ralph 2006). The relationship between the items (not just the items independently) would be a useful addition to Table 2, giving another reflection of demands of the task.

-Finally, older people are more accurate but slower across all tasks, although the degree to which they are varies (sometimes significantly). This suggests perhaps that they are more considered in their response or less familiar with cognitive testing. Did you consider computing response efficiency, or finding some other way to capture this trade off?

Decision letter (RSOS-211056.R0)

Dear Dr Wu

On behalf of the Editors, we are pleased to inform you that your Manuscript RSOS-211056 "Validated measures of semantic knowledge and semantic control: Normative data from young and older adults for more than 300 semantic judgements" has been accepted for publication in Royal Society Open Science subject to minor revision in accordance with the referees' reports. Please find the referees' comments along with any feedback from the Editors below my signature.

Please submit your revised manuscript and required files (see below) no later than 7 days from today's (ie 25-Nov-2021) date. Note: the ScholarOne system will 'lock' if submission of the revision is attempted 7 or more days after the deadline. If you do not think you will be able to meet this deadline please contact the editorial office immediately.

on behalf of Dr Emma Hayiou-Thomas (Associate Editor) and Essi Viding (Subject Editor)
openscience@royalsociety.org

Associate Editor Comments to Author (Dr Emma Hayiou-Thomas):

Associate Editor: 1

Comments to the Author:

Thank you for this interesting and very clearly presented paper, which I enjoyed reading! As you will see, the reviewers had some very positive things to say, but there were also some substantial concerns which will need to be addressed in any revision. The most serious of these relates to the potential involvement of semantic control processes in your semantic knowledge task; specifically, that the strength of the associations within word-pairs do not appear to have been experimentally controlled in this task, thus resulting in a potential confound with semantic control. Given the importance of the distinction between semantic knowledge and control for

your study, please make sure to provide further information on this aspect of the stimuli, and discuss the implications for interpreting your results. The reviewers have made a number of additional thoughtful points and constructive suggestions which will need to be responded to. I look forward to seeing a revised version of this manuscript, and would like to reiterate my apologies for the length of time it has taken to get back to you with a decision.

Reviewer comments to Author:

Reviewer: 1

Comments to the Author(s)

Thank you for the opportunity to review this study, which makes an important and very useful contribution to the field of semantic cognition, across populations. I am sure that many research groups will benefit from the use of the norms provided by this study! This study administered a semantic knowledge (synonym judgment task) and a semantic control (feature-matching task) to younger (n=127) and older (n=106) adults and replicated previous findings that older adults tend to have a deeper semantic store, yet are out-performed by their younger counterparts on tasks that require efficient control over these representations. It also produces a set of normed data and demonstrates the relationship between TDS and performance (i.e., lower performance for low-TDS items), and how this effect is amplified in older individuals. I only have a few very minor comments below.

I noticed that your young age range is 18-42, however, based on your mean and SD, it looks like very few participants will be over 25. How many participants are 'outliers' in terms of age? Do your results change/is their behaviour different to the rest of the younger group? (I know the age range is large for the older age group – but this is more standard in the literature. It would still be interesting to see whether age has an effect within this group.)

Could you possibly provide information about proportion of RT's winsorized? For example, are there any trials that consistently needed winsorizing across participants?

Given the older adults performance on the semantic control and cognitive control tasks is correlated, but not the semantic control semantic knowledge; what implications does this have for the processes engaged to perform the semantic control task (i.e., feature matching) you use in this study? (e.g., does this task engage semantic control or executive control?) Maybe I missed this, but did you compare your synonym judgment task to other standardized assessments of semantic knowledge?

Do you have any citations to support your notion that a time-limit might encourage older people to respond more quickly at the cost of accuracy?

Reviewer: 2

Comments to the Author(s)

This manuscript presents norms from 127 young and 106 older participants for two semantic tasks. The authors sustain that these tasks assess the organization of knowledge in the semantic store (synonym judgment task) and control processes involved in the retrieval of this knowledge (feature-matching task). They also affirm that their results are valid because they corroborate previous age- and linguistic-related effects.

In my view the main problem of this study is that it is described as a normative study. I find the experimental methodology and the analytical approach not solid enough for the aim of the study (i.e., building norms from young and older participants for semantic knowledge and control tasks). Moreover, I'm not sure that the tasks actually assess what the authors aim to assess. For

these reasons, I will focus my review on the major critical points of the study. I am sorry I cannot be more positive but I hope that my comments can be helpful and constructive.

- 1) The authors write that the tasks dissociate the level of semantic knowledge and control processes, but I don't think this is the case. Indeed, the fact that the authors did not measure the relation between cue and target (in terms of association strength, lexical co-occurrence for example) in the synonym judgement doesn't mean that this relationship does not have an effect on participants' performance and that the retrieval of these links do not require control. It is reasonable to think that some word pairs are more associated (or more similar in meaning) compared to others, so for these latter, the controlled retrieval is needed to access to them. At some point in the Introduction the authors themselves write that the semantic control requires "flexible retrieval of less salient aspects of knowledge" (i.e., controlled retrieval) (pag. 5). For this reason, I would recommend the authors to measure the link between word pairs in the semantic judgments and avoid the division between semantic knowledge and control processes. Perhaps, it would be preferable to talk about semantic processing in general given the nature of the tasks.
- 2) Related to my previous point, in many tasks (like the synonym judgment task) requiring the processing of semantic information is really difficult to understand if the effects found are due to the manipulation of information at the representation level or manipulation of information for its retrieval. For example, how can the authors exclude that the word frequency effect found in the synonym judgement is due to the greater control demands required for the retrieval of low frequent words?
- 3) The tasks included in this study include only a small number of stimuli. To function as a norm, this study should have included a very large number of stimuli allowing researchers to select and manipulate word variables of interest (see for example, the Semantic Priming Project or other norms with linguistic stimuli, like Small World of Words for English language, and so forth).
- 4) Another issue I see is that the tasks should have been controlled for their general difficulty, especially because they assess the capacity of cognitive control demands (specific to the semantic domain and more general). This is also evident when we consider the semantic tasks only. Indeed, the authors write that "The items used in the semantic knowledge test were significantly lower in frequency [...], concreteness [...], and summed word length [...] than those used in the semantic control test." (pag. 9). More importantly, this difference among tasks emerged in the results (pag. 13, see main effect of test on RT and from the Table 6). I think this is a great issue in this study, especially given the dispute in literature about how much executive functions and semantic control overlap in terms of cognitive processes and brain underpinnings.
- 5) I wonder why the response mapping was different between the synonym judgement and the feature-matching task (pag. 10).
- 6) It is unclear to me how the authors built the linear mixed effect models. For example, were the confounding variables (at least the ones for which the semantic tasks differed) included in the analyses?
- 7) Pag. 14, line 14-22: this paragraph is more part of a Discussion section than a Results section.
- 8) There is no reliability measure of the data (i.e., split-half correlation). I would recommend the authors to perform these analyses to control if the data are consistent across participants and items.

Reviewer: 3

Comments to the Author(s)

This is a detailed battery, with analysis of a significant dataset, with openly available stimuli and data, warranting publication. The main element which could be further addressed is describing the stimuli in greater detail, particularly considering the task difficulty manipulation, as described below.

-The semantic knowledge task has a difficulty manipulation in terms of the frequency of words. The distinction with semantic control could be further addressed – e.g., are the words in the low

frequency manipulation as related (in semantic space) as high frequency words? Semantic distance has previously been used as a difficulty manipulation (Noonan et al., 2010) and found to relate to semantic control deficits, so it would be worth including this detail. If semantic distance could be shown to be the same for high vs. low frequency words, the argument may be more compelling. It also seems that some literature regarding frequency and its relationship to control could be added (the reduced/ reversed frequency effect in SA - and papers on synonym judgement in SA).

-A further consideration to this end might be to consider co-occurrence (or frequency) of the two items combined (as previously explored, e.g., Jefferies & Lambon Ralph 2006). The relationship between the items (not just the items independently) would be a useful addition to Table 2, giving another reflection of demands of the task.

-Finally, older people are more accurate but slower across all tasks, although the degree to which they are varies (sometimes significantly). This suggests perhaps that they are more considered in their response or less familiar with cognitive testing. Did you consider computing response efficiency, or finding some other way to capture this trade off?

===PREPARING YOUR MANUSCRIPT===

one version should clearly identify all the changes that have been made (for instance, in coloured highlight, in bold text, or tracked changes);

===PREPARING YOUR REVISION IN SCHOLARONE===

-- If you are requesting an article processing charge waiver, you must select the relevant waiver option (if requesting a discretionary waiver, the form should have been uploaded, see 'File upload' above).

-- If you have uploaded any electronic supplementary (ESM) files, please ensure you follow the guidance at <https://royalsociety.org/journals/authors/author-guidelines/#supplementary-material> to include a suitable title and informative caption. An example of appropriate titling and captioning may be found at https://figshare.com/articles/Table_S2_from_Is_there_a_trade-off_between_peak_performance_and_performance_breadth_across_temperatures_for_aerobic_scope_in_teleost_fishes_/3843624.

Author's Response to Decision Letter for (RSOS-211056.R0)

See Appendix A.

Decision letter (RSOS-211056.R1)

Dear Dr Wu,

It is a pleasure to accept your manuscript entitled "Validated measures of semantic knowledge and semantic control: Normative data from young and older adults for more than 300 semantic judgements" in its current form for publication in Royal Society Open Science. The comments of the reviewer(s) who reviewed your manuscript are included at the foot of this letter.

on behalf of Dr Emma Hayiou-Thomas (Associate Editor) and Essi Viding (Subject Editor)
openscience@royalsociety.org

Appendix A

Reviewer comments to Author:

Reviewer: 1

Comments to the Author(s)

Thank you for the opportunity to review this study, which makes an important and very useful contribution to the field of semantic cognition, across populations. I am sure that many research groups will benefit from the use of the norms provided by this study! This study administered a semantic knowledge (synonym judgment task) and a semantic control (feature-matching task) to younger (n=127) and older (n=106) adults and replicated previous findings that older adults tend to have a deeper semantic store, yet are out-performed by their younger counterparts on tasks that require efficient control over these representations. It also produces a set of normed data and demonstrates the relationship between TDS and performance (i.e., lower performance for low-TDS items), and how this effect is amplified in older individuals. I only have a few very minor comments below.

****Response: We thank the reviewer for their thoughtful and encouraging comments.**

I noticed that your young age range is 18-42, however, based on your mean and SD, it looks like very few participants will be over 25. How many participants are 'outliers' in terms of age? Do your results change/is their behaviour different to the rest of the younger group?

(I know the age range is large for the older age group – but this is more standard in the literature. It would still be interesting to see whether age has an effect within this group.)

****Response: We found that there were only 5 participants in the young group who were older than 25. We have recalculated our major mixed effects models after excluding the data of the 5 "outliers" (including the model evaluating age groups*tests effects, the model probing frequency*age groups effects in knowledge task, and the model probing TDS*age groups effects in semantic control task). None of the significance of the effects was changed.**

Could you possibly provide information about proportion of RT's winsorized? For example, are there any trials that consistently needed winsorizing across participants?

****Response: We found that only a small proportion of RTs were winsorized in our study (i.e., 4.9%). We have added this piece of information in the Methods.**

Given the older adults performance on the semantic control and cognitive control tasks is correlated, but not the semantic control semantic knowledge; what implications does this have for the processes engaged to perform the semantic control task (i.e., feature matching) you use in this study? (e.g., does this task engage semantic control or executive control?)

Maybe I missed this, but did you compare your synonym judgment task to other standardized assessments of semantic knowledge?

****Response: This is a good point. We have added some more explanation in the Discussion to discuss the relationship/correlation between the two control tasks. Because we didn't collect data using other knowledge tasks, we cannot compare our knowledge task with other standardized assessments of semantic knowledge with the current dataset.**

Do you have any citations to support your notion that a time-limit might encourage older people to respond

more quickly at the cost of accuracy?

****Response: We have referenced the relevant paper in the Discussion.**

Reviewer: 2

Comments to the Author(s)

This manuscript presents norms from 127 young and 106 older participants for two semantic tasks. The authors sustain that these tasks assess the organization of knowledge in the semantic store (synonym judgement task) and control processes involved in the retrieval of this knowledge (feature-matching task). They also affirm that their results are valid because they corroborate previous age- and linguistic-related effects.

In my view the main problem of this study is that it is described as a normative study. I find the experimental methodology and the analytical approach not solid enough for the aim of the study (i.e., building norms from young and older participants for semantic knowledge and control tasks). Moreover, I'm not sure that the tasks actually assess what the authors aim to assess. For these reasons, I will focus my review on the major critical points of the study. I am sorry I cannot be more positive but I hope that my comments can be helpful and constructive.

****Response: We thank the reviewer for their thoughtful and encouraging comments.**

1) The authors write that the tasks dissociate the level of semantic knowledge and control processes, but I don't think this is the case. Indeed, the fact that the authors did not measure the relation between cue and target (in terms of association strength, lexical co-occurrence for example) in the synonym judgement doesn't mean that this relationship does not have an effect on participants' performance and that the retrieval of these links do not require control. It is reasonable to think that some word pairs are more associated (or more similar in meaning) compared to others, so for these latter, the controlled retrieval is needed to access to them. At some point in the Introduction the authors themselves write that the semantic control requires "flexible retrieval of less salient aspects of knowledge" (i.e., controlled retrieval) (pag. 5). For this reason, I would recommend the authors to measure the link between word pairs in the semantic judgments and avoid the division between semantic knowledge and control processes. Perhaps, it would be preferable to talk about semantic processing in general given the nature of the tasks.

****Response: To address the reviewer's point, we have added information on the semantic relatedness of the word pairs in the synonym task to Table 2 (in the TDS row). We have also expanded the introduction to provide greater explanation on the issue mentioned in this comment. In short, no task is a pure measure of semantic knowledge or semantic control, but different tasks can emphasize different aspects of semantic cognition. Lastly, by calculating the correlations between difficulty levels and frequency/TDS of the word pairs in the synonym task, we found that the difficulty levels in the synonym task (which were generated based on behavioral performance) were highly correlated with frequency ($r = -0.561$) but not the TDS information ($r = -0.006$; data added to results section). The reverse was true for the semantic control task. This provides further evidence that the relatedness of the word pairs is not a major determinant of performance in the synonym task.**

2) Related to my previous point, in many tasks (like the synonym judgment task) requiring the processing of semantic information is really difficult to understand if the effects found are due to the manipulation of information at the representation level or manipulation of information for its retrieval. For example, how can the authors exclude that the word frequency effect found in the synonym judgement is due to the greater control demands required for the retrieval of low frequent words?

****Response:** In fact, as we now note in the paper, previous studies have shown that semantic control demands are not generally greater for lower frequency words and that sometimes the reverse is true (Almaghyuli et al., 2012; Hoffman et al., 2011). Because high-frequency words tend to appear in a broader range of linguistic contexts and have more variable meanings (i.e., more semantically diverse), they exert greater demands on cognitive control.

3) The tasks included in this study include only a small number of stimuli. To function as a norm, this study should have included a very large number of stimuli allowing researchers to select and manipulate word variables of interest (see for example, the Semantic Priming Project or other norms with linguistic stimuli, like Small World of Words for English language, and so forth).

****Response:** We agree that some linguistic norms include a larger number of stimuli. However, unlike those norms, our norms required more delicate stimulus selection which limited the sample size of our stimuli. For example, in each trial of the feature-matching task, we had to match words based on specific features and cover a wide range of TDS between the words at the same time. Besides, it is not uncommon for assessments of semantics to have a smaller number of items (e.g., the Mill Hill Vocabulary Scale).

4) Another issue I see is that the tasks should have been controlled for their general difficulty, especially because they assess the capacity of cognitive control demands (specific to the semantic domain and more general). This is also evident when we consider the semantic tasks only. Indeed, the authors write that “The items used in the semantic knowledge test were significantly lower in frequency [...], concreteness [...], and summed word length [...] than those used in the semantic control test.” (pag. 9). More importantly, this difference among tasks emerged in the results (pag. 13, see main effect of test on RT and from the Table 6). I think this is a great issue in this study, especially given the dispute in literature about how much executive functions and semantic control overlap in terms of cognitive processes and brain underpinnings.

****Response:** Indeed, as reported in the current paper, our tests were different in some properties. This was caused by the nature of the experimental paradigms and the methods we used to select stimuli. It is important to note, however, that our study aimed to compare the performance of different age groups in each task, but not to compare absolute levels of performance on different tasks. The replication analyses of existing psycholinguistic-property effects were also conducted within individual tasks (e.g., frequency effect in the knowledge task), but not across tasks.

5) I wonder why the response mapping was different between the synonym judgement and the feature-matching task (pag. 10).

****Response:** we used ‘s’ and ‘d’ as the response buttons to stand for ‘similar’ and ‘different’ word pairs in the synonym task, which was more straightforward and could avoid potential confusion caused by using ‘1’ and ‘2’.

6) It is unclear to me how the authors built the linear mixed effect models. For example, were the confounding variables (at least the ones for which the semantic tasks differed) included in the analyses?

****Response:** Predictors in the model testing general performance were age group and test type. Predictors in the model probing frequency effect were age group, frequency and item type (i.e., similar- and dissimilar-word-pair items). Predictors in the model probing TDS effect were age group and TDS. We also included the trial order in experiment in each model as a covariate of no interest. We have ensured this information is included in our Methods.

7) Pag. 14, line 14-22: this paragraph is more part of a Discussion section than a Results section.

****Response: Because these sentences provide important context for interpreting this specific part of the results, we think it is better to put it in this section.**

8) There is no reliability measure of the data (i.e., split-half correlation). I would recommend the authors to perform these analyses to control if the data are consistent across participants and items.

****Response: Thanks for this suggestion – we have added a reliability analysis in our paper. We found a mean split-half correlation of $r > 0.81$ in each task.**

Reviewer: 3

Comments to the Author(s)

This is a detailed battery, with analysis of a significant dataset, with openly available stimuli and data, warranting publication. The main element which could be further addressed is describing the stimuli in greater detail, particularly considering the task difficulty manipulation, as described below.

****Response: We thank the reviewer for their thoughtful and encouraging comments.**

-The semantic knowledge task has a difficulty manipulation in terms of the frequency of words. The distinction with semantic control could be further addressed – e.g., are the words in the low frequency manipulation as related (in semantic space) as high frequency words? Semantic distance has previously been used as a difficulty manipulation (Noonan et al., 2010) and found to relate to semantic control deficits, so it would be worth including this detail. If semantic distance could be shown to be the same for high vs. low frequency words, the argument may be more compelling. It also seems that some literature regarding frequency and its relationship to control could be added (the reduced/ reversed frequency effect in SA - and papers on synonym judgement in SA).

****Response: This is a good point. To check whether there was a relationship between frequency and TDS (i.e., semantic distance) of the word pairs in the knowledge task, we calculated the correlation between frequency and TDS. The results revealed no significant relationship between these two variables ($r = 0.073$, $p = 0.362$). Regarding the second half of this comment, we have referenced the relevant studies in the paper.**

-A further consideration to this end might be to consider co-occurrence (or frequency) of the two items combined (as previously explored, e.g., Jefferies & Lambon Ralph 2006). The relationship between the items (not just the items independently) would be a useful addition to Table 2, giving another reflection of demands of the task.

****Response: We have added the TDS information (i.e., co-occurrence) of the word pairs in the synonym task to Table 2.**

-Finally, older people are more accurate but slower across all tasks, although the degree to which they are varies (sometimes significantly). This suggests perhaps that they are more considered in their response or less familiar with cognitive testing. Did you consider computing response efficiency, or finding some other way to capture

this trade off?

****Response: We are grateful for this suggestion. However, response efficiency measures can't be used in trial-level mixed effects analysis, which is our preferred approach for modelling the data.**

Almaghyuli, A., Thompson, H., Ralph, M. A. L., & Jefferies, E. (2012). Deficits of semantic control produce absent or reverse frequency effects in comprehension: evidence from neuropsychology and dual task methodology. *Neuropsychologia*, *50*(8), 1968-1979.

Hoffman, P., Rogers, T. T., & Ralph, M. A. L. (2011). Semantic diversity accounts for the "missing" word frequency effect in stroke aphasia: Insights using a novel method to quantify contextual variability in meaning. *Journal of Cognitive Neuroscience*, *23*(9), 2432-2446.